# Integration of Hepatitis C and Addiction Treatment in People Who Inject Drugs: The San Patrignano HCV-Free and Drug-Free Experience

**DOI:** 10.3390/v16030375

**Published:** 2024-02-28

**Authors:** Pierluca Piselli, Antonio Boschini, Romina Gianfreda, Alessandra Nappo, Claudia Cimaglia, Gianpaolo Scarfò, Camillo Smacchia, Raffaella Paoletti, Sarah Duehren, Enrico Girardi

**Affiliations:** 1Clinical Epidemiology Unit, Department of Epidemiology and Preclinical Research, National Institute for Infectious Diseases Lazzaro Spallanzani IRCCS, 00149 Rome, Italy; alessandra.nappo@inmi.it (A.N.); claudia.cimaglia@inmi.it (C.C.); gianpaolo.scarfo@inmi.it (G.S.); enrico.girardi@inmi.it (E.G.); 2Medical Center, Comunità di S. Patrignano, 47853 Coriano, RN, Italy; aboschini@sanpatrignano.org (A.B.); camillo.smacchia@aulss9.veneto.it (C.S.); raffaellapoletti8@gmail.com (R.P.); sduehren@alumni.nd.edu (S.D.); 3Infectious Disease Unit, Hospital of Rimini “Gli Infermi”, 47923 Rimini, Italy; romina.gianfreda@auslromagna.it

**Keywords:** HCV, cascade of care, therapeutic community, intravenous drug use

## Abstract

Injection drug use represents an important contributor to hepatitis C virus (HCV) transmission, hence therapeutic communities (TCs) are promising points of care for the identification and treatment of HCV-infected persons who inject drugs (PWIDs). We evaluated the effectiveness and efficacy of an HCV micro-elimination program targeting PWIDs in the context of a drug-free TC; we applied the cascade of care (CoC) evaluation by calculating frequencies of infection diagnosis, confirmation, treatment and achievement of a sustained virological response (SVR). We also evaluated the risk of reinfection of PWIDs achieving HCV eradication by collecting follow-up virologic information of previously recovered individuals and eventual relapse in drug use, assuming the latter as a potential source of reinfection. We considered 811 PWIDs (aged 18+ years) residing in San Patrignano TC at the beginning of the observation period (January 2018–March 2022) or admitted thereafter, assessing for HCV and HIV serology and viral load by standard laboratory procedures. Ongoing infections were treated with direct-acting antivirals (DAA), according to the current national guidelines. Out of the 792 individuals tested on admission, 503 (63.5%) were found to be seropositive for antibodies against HCV. A total of 481 of these 503 individuals (95.6%) underwent HCV RNA testing. Out of the 331 participants positive for HCV RNA, 225 were ultimately prescribed a DAA treatment with a sustained viral response (SVR), which was achieved by 222 PWIDs (98.7%). Of the 222 PWIDs, 186 (83.8%) with SVR remained HCV-free on follow-up (with a median follow-up of 2.73 years after SVR ascertainment). The CoC model in our TC proved efficient in implementing HCV micro-elimination, as well as in preventing reinfection and promoting retention in the care of individuals, which aligns with the therapeutic goals of addiction treatment.

## 1. Introduction

People who inject drugs (PWIDs) are responsible for half of the 1.7 million new cases of hepatitis C virus (HCV) transmissions occurring worldwide every year [1]. While this population is the most exposed to HCV infection and transmission [2], it is also susceptible to stigma, socioeconomic problems, and poor access to healthcare. Altogether, these constraints, and the high risk of reinfection after HCV cure, make HCV elimination a difficult goal to achieve in this population, even in the era of effective and user-friendly direct-acting antivirals (DAA) against HCV [2,3].

The cascade of care (CoC) has emerged as an efficient system to evaluate programs for the diagnosis, linkage to care, and treatment of HCV [4]. By dividing HCV treatment into a series of well-defined steps necessary for achieving a sustained virologic response (SVR), the care cascade allows for consistent monitoring of global progress toward public health goals and facilitates the identification of critical areas of care delivery systems that may need improvement. 

Population-based estimates of the care cascade for HCV among PWIDs show very low levels of treatment engagement, with only 8–17% beginning DAA therapy and 7% completing their course [5,6,7].

To overcome this problem, international guidelines recommend that HCV treatment in drug users should be integrated with treatment for drug addiction [8]. Implementing HCV treatment in institutions outside of the traditional confines of the healthcare system, lowering barriers to healthcare access, and reducing stigma may facilitate successful progression through the cascade of care [9]. These objectives can be reached through two major approaches to drug addiction treatment, harm reduction and recovery-oriented programs. 

Harm reduction does not directly address addiction, but aims to reduce health, social, and legal consequences of substance use disorders (SUD), including infections such as HCV, drug overdoses, crime, prostitution, and social marginalization [10]. The tools of this public health approach are opioid agonist treatment (OAT), needle exchange programs (NEP), supervised injection sites, low-threshold services, mobile outreach programs, etc. In the harm reduction community-based setting, the rate of successful HCV cure (SVR) in active drug users results improved (42% among those eligible for treatment and 89.6% among those completing DAA treatment) [11]. However, the main limitation of this approach is that the problem of addiction is not addressed as a priority, and there is still a risk of HCV reinfection [12].

On the other hand, recovery-oriented treatments pursue the aim of recovery, defined as “a voluntarily maintained lifestyle characterized by abstinence, personal health, and citizenship” [13]. Among recovery models, there are different approaches: outpatient organizations such as “Narcotics Anonymous”, short-term residential treatment (“Rehab”) and long-term residential treatment (therapeutic communities, TCs). In this study, we will discuss HCV micro-elimination in a therapeutic community treatment setting. The major drawbacks of therapeutic communities, especially those that do not treat with opioid agonists, is the low rate of retention in treatment (9% to 56%) [14], due to the length of treatment, the intensity of the program and the need for long-lasting motivation to fully engage with the program. Long-term recovery from addiction is directly related to the length of stay in residential treatment [15,16]; however, despite a long research tradition in TCs [17,18], the evidence base for the effectiveness of TCs is limited, according to the prevailing Cochrane hierarchy of scientific evidence [19].

To our knowledge, no published data exist regarding HCV therapy in people with non-pharmacological SUD treatment living in therapeutic communities. There are studies on the micro-elimination of HCV in prisons [20,21,22], but this context is different from that of a TC for several reasons, particularly the risk of dropout. 

Our study describes the CoC of HCV treatment in the TC of San Patrignano, the largest TC in Europe [23]. 

The main objective was to verify the feasibility, effectiveness, and any strengths and weaknesses in the different phases of the cascade of care. A second objective was to assess the risk of reinfection in a population in which the treatment of hepatitis C is supplemented with SUD treatment aimed at recovery.

## 2. Materials and Methods

### 2.1. Study Setting 

The study was conducted in the community of ‘‘San Patrignano’’ (SPTC), a private residential community for the rehabilitation of people affected by SUDs, located in Northern Italy [24]. The property, which extends for 1000 acres, consists of several residential buildings and extensive pastoral and agricultural lands. San Patrignano has welcomed, in the past 42 years, over 26,000 people, offering them a home, health and legal assistance, the opportunity to study, learn a job and ultimately change their life [23].

The treatment, which lasts about 30 months, is free of charge. Opioid agonist treatment (OAT) is used only in the detoxification phase upon entry into San Patrignano. 

Individuals can have access to SPTC through (a) public health services (SerD), (b) a network of non-profit associations all over Italy and other European countries or (c) through the justice system, frequently as alternative sentencing. 

### 2.2. Study Population 

Among roughly 2400 subjects already residing at SPTC at the start of the study (1 January 2018) or admitted thereafter (until 31 March 2022), we considered in this study cohort only PWIDs currently in treatment for addiction who were 18 years or older. 

Baseline information was recorded for each individual on admission, including demographic data, date of admission to SPTC and duration of stay, substance use history (age at first use of injectable substances and duration of addiction, substances abused, history of OAT), history of incarceration and history of HCV testing and treatment.

Serologic testing for HCV and HIV is routinely offered to residents, and clinical and laboratory data are routinely collected by the treating physicians. The Medical Center, including outpatient facilities and a 50-bed ward, provides care and treatment to all residents [25,26,27]. This study was approved by the ethics committee of the local health authority (Area Vasta Romagna, CEROM ref n. 3373/2019).

### 2.3. Assessment of HCV Infection

HCV testing was performed using standard laboratory procedures at Rimini Hospital. Briefly, blood samples were collected from each participant at the time of enrolment (at baseline). All the PWIDs found to be seropositive for HCV were tested for the presence of HCV RNA. Viraemic subjects were eligible for treatment with DAA. Anti-HCV seropositive persons who were HCV RNA-negative were classified as having spontaneously cleared the infection or been cured with previous HCV treatment, depending on the availability of the treatment history. Individuals with chronic HCV infection were assessed for liver fibrosis using a FibroScan procedure and staged according to the METAVIR fibrosis scoring system, and classified as absence of fibrosis (F0), mild fibrosis (F1—portal fibrosis), significant fibrosis (F2—periportal fibrosis), severe fibrosis (F3—bridging fibrosis) or presence of cirrhosis (F4) [28].

The data on anti-HCV antibodies, ribonucleic acid testing, treatment and its outcome were used to populate the HCV care cascade.

For the cascade of care (CoC), we assessed the transition through the HCV care cascade by extending consecutive cascade milestones previously used [29]. In the current study, the stages of the HCV care cascade were defined as 1. testing for HCV antibodies; 2. reactivity to the HCV antibody test; 3. access to HCV RNA testing; 4. HCV RNA confirmation; 5. DAA treatment initiation; and 6. treatment completion and post-treatment assessment for HCV cure. All the stages of HCV care were carried out inside the SPTC Medical Center.

DAA treatment was prescribed according to the standard Italian guidelines (8 or 12 weeks of treatment with DAA) and administered during the residential TC program; adherence was supervised by trained caregivers within the TC. The effect of HCV treatment was assessed at the end of treatment, and the virological response (HCV RNA-negative) at this time point was defined as “end of treatment response” (EoTR). An HCV cure was defined as a sustained virological response (SVR)—being negative for HCV RNA at 12 weeks after the end of treatment.

PWIDs leaving SPTC at the end of their treatment for addiction were considered discharged, while those who voluntarily quit before completing their addiction treatment period were considered dropouts. 

The follow-up for HCV reinfection was conducted (a) in SPTC Medical Center, for those still in treatment or followed as outpatients; (b) by contacting the person directly for those discharged from SPTC; and (c) through the above-mentioned network of non-profit associations. Follow-up was completed on 31 December 2022.

### 2.4. Statistical Analysis

The statistical analysis involved using descriptive statistics by reporting frequencies and percentages for qualitative variables, while the median and interquartile range (IQR) were used to describe quantitative variables. 

A comparison of study participants’ characteristics was performed using the chi-square test for trend, chi-square or Fisher exact tests for categorical variables, and the Mann–Whitney test for numeric variables, as appropriate.

Milestones of HCV CoC achievement, as defined above, were reported as counts and relative percentages. 

CoC from study enrolment to DAA response was reported using a flow chart, which included reasons for no treatment, treatment interruption or incomplete assessments.

Moreover, a Poisson regression analysis was performed to identify factors associated with DAA treatment initiation in chronically HCV-infected individuals not already treated with DAA, present at SPTC on 1 January 2018 or entered thereafter. This analysis included computing incidence rate ratios (IRRs) and corresponding 95% confidence intervals (CI), both in univariable and multivariable regression (then calculating adjusted IRR aIRR) [30]. Person-days (PDs) at risk for DAA treatment initiation were calculated from 1 January 2018, or the date of entry to SPTC (for those entered after 1 January 2018), to the date of DAA initiation, last follow-up visit, date of death, or the end date of the study (31 March 2022), whichever came first.

All the statistical analyses were performed using SPSS Statistical Software ver. 28.0 (IBM SPSS Statistics, IBM Corp., Armonk, NY, USA) or STATA ver. 17 (StataCorp LLC, College Station, TX, USA), and a *p*-value < 0.05 was considered statistically significant.

## 3. Results

### 3.1. Baseline Characteristics

A total of 811 PWIDs admitted to SPTC during the study period were included and evaluated. There were 625 males (77.1%), with a median age of 32 years at entry (IQR: 26–39), who were mostly Italians (92.6%) (Table 1). The median age at initiation of intravenous drug use was 20 years (IQR: 18–24) with a median period of drug use of 7 years (IQR: 3–16). The vast majority of PWIDs reported to be addicted to heroin (97.2%) with or without cocaine or other substances. In addition, 84.1% of the PWIDs had been on OAT treatment (maintenance or tapering) before SPTC entry. 

Almost a quarter of PWIDs had a history of incarceration (26.6%), and 55 (6.8%) tested positive for HIV. Regarding their history of previous HCV testing, 642 (79.2%) individuals had been tested for HCV before SPTC entry, of which 377 (58.7%) reported an HCV-positive test result. Out of these 377, 82 (21.8%) were previously treated for HCV infection with either interferon/ribavirin or DAA, and 64 (78.0%) reported a virological response (SVR).

### 3.2. HCV-Positivity Assessment and HCV Cascade of Care

A vast majority of participants (792, 97.7%) were screened for HCV infection at entry, while only a small fraction of PWIDs (n = 19) were not tested due to early dropping out of SPTC. 

A total of 503 PWIDs (63.5%) were found to be anti-HCV seropositive, of which 377 were aware of their infection status (74.9%). Out of 503 PWIDs found HCV-positive, 35 had a history of a most recent HCV-negative test, and 91 had never been tested before for HCV; this latter group corresponded to a seroprevalence of 58.3% among those never tested for HCV before entry (n = 156). Table 2 reports all the characteristics of PWIDs tested for HCV Ab categorized according to HCV positivity.

HCV seropositivity was associated with older age compared to HCV-free individuals (median age 35 years vs. 29, *p* < 0.001). HCV seropositivity was also associated with a longer period of drug addiction (11 vs. 4 years, *p* < 0.001) and a younger age at initiation of intravenous drug use (20 vs. 21 years, *p* < 0.001), compared to HCV-free individuals. HCV-seropositive PWIDs were more likely to be co-infected with HIV (*p* < 0.001) and have a history of incarceration (*p* < 0.001). No association with gender or nationality was observed. 

Among those found to be seropositive for HCV, HCV RNA testing was completed among 481/503 individuals (95.6%); 22 subjects were not tested for HCV RNA because of being discharged early (n = 4) or dropping out (n = 18) (Figure 1).

The proportion of those found to be HCV RNA positive was 68.8% (331/481), and all of them were considered and proposed for DAA treatment. Out of 331 subjects with positive HCV RNA, 326 (98.5%) were sequenced. Genotypes 1 and 3 were the most prevalent (164—50.3% and 134—41.1%, respectively), with predominant subtype 1a (143/164, 87.2%) and 3a (126/134, 94.0%) prevalence.

Among these 331 viremic subjects, 311 were treatment-naïve while 20 were considered as re-infection, given a documented SVR from a previous anti-HCV treatment (out of 64 persons above mentioned).

Overall, during the study period, DAA was prescribed to 225 (68.0%) chronically infected subjects, while 106 (32.0%) HCV RNA-positive PWIDs did not receive DAA therapy due to either dropping out (n = 42—39.6%) or being discharged from SPTC before the start of DAA treatment (n = 62—58.5%). Moreover, two enlisted subjects were not treated with DAA due to worsening health conditions, not related to HCV infection, which precluded them from DAA initiation.

Figure 1 schematically reports the path of all the PWIDs included in the study to calculate the final CoC.

The results of the analysis conducted to identify factors associated with DAA initiation are shown in Table 3.

In the univariate analysis, a diagnosis of severe fibrosis or cirrhosis (Metavir score F3–F4), HIV positivity, being Italian and having a late SPTC admission were associated with a faster DAA initiation, while no association was found according to age and gender. 

In the final model, factors significantly associated with DAA initiation were liver disease severity by Fibroscan (aIRR = 2.43 for those with F3–F4 and aIRR = 1.58 for those with F2 vs. those with F0-F1, *p* < 0.001), being HIV-positive (aIRR = 2.39, *p* < 0.001) and a late period of last entry to the SPTC (aIRR = 4.25 for those last admitted in 2020 or later and aIRR = 1.48 for those admitted within 2018–2019 vs. those admitted before 2018, *p* < 0.001). Moreover, non-Italians were least likely to be treated (aIRR = 0.60, *p* < 0.080) while, on the contrary, HIV-positive persons were more likely to be treated (aIRR = 2.39).

In terms of the results of treatment, among 225 PWIDs who started treatment with DAA, three subjects (1.3%) completed DAA treatment and reached EoTR but left SPTC before SVR assessment (two dropouts and one subject discharged before assessment), whereas 222 (98.7%) PWIDs were completely assessed and considered cured (SVR12). Figure 2 summarizes the achievement of the milestone of CoC as above defined.

### 3.3. Outcome and HCV Reinfection

Follow-up data were actively collected on all 222 patients successfully treated, with a median of 2.73 years (IQR: 2.42–3.52 years) post-SVR ascertainment. Of these, 134 (60.4%) underwent follow-up testing for HCV RNA (83 still at SPTC), and no cases of HCV RNA infection were observed (Figure 3).

No virological follow-up data was available for the remaining 88 PWIDs (72 regularly discharged and 16 who dropped out). Of these 88 PWIDs, 53 subjects who were contacted at follow-up remained abstinent from drugs, and we can assume they were unlikely to be HCV-reinfected. Twenty-six PWIDs relapsed on drugs (twenty-one regularly discharged and five dropped out), while for the remaining nine (three discharged and six dropped out), data on eventual relapse on drugs was missing. 

Overall, among 222 successfully treated PWIDs, 187 (84.2%) can be considered not reinfected during follow-up because they were found to be HCV RNA-negative at follow-up re-testing (134/187 = 71.7%) or because they remained abstinent from drugs (53/187, 28.3%). Potentially, HCV reinfection could have occurred in up to 35 subjects who relapsed on drugs or for which information on HCV RNA testing or drug relapse is missing.

## 4. Discussion

Despite the availability of effective and user-friendly direct-acting antiviral drugs (DAAs), the eradication of HCV in the PWID population is a difficult goal to achieve due to various factors, including stigma, limited access to care, and the risk of reinfection after effective treatment.

It is a shared opinion that the treatment of HCV infection should be combined with the treatment of SUD and that one of the most effective strategies is to design micro-elimination interventions. These interventions differ depending on the varying contexts in which they operate but are united with the goal of bringing health care “on the spot” by concentrating all the phases that lead to improved access to care (“cascade of care”) in a single area (“point of care”), rendering it easier to meet patients [31,32]. This avoids the need for patients to access other healthcare facilities, which may sometimes be a place of stigma, deprivation of privacy, or bureaucratic delays.

This study describes an experience of the micro-elimination of hepatitis C in a population that, as far as we know, has never been demonstrated in people recovering from drug addiction in a residential TC (San Patrignano, Italy). 

TCs are one of the therapeutic responses to SUD; the first of these types of communities were born in the US during the 1960s and then further developed throughout the world, differentiating themselves significantly according to their cultural contexts. These TCs aim at recovery, defined as “a voluntarily maintained lifestyle characterized by abstinence, personal health and citizenship” that can be reached with or without (as in San Patrignano TC) the use of agonist medications. [13].

TCs are almost always born from private initiatives and are more often connected to spiritual movements than to scientific research centers. Historically, there has not been a culture of data collection and publication within these TCs. Therefore, their treatment models were not considered “evidence-based”.

There is little literature on their effectiveness in terms of retention in treatment (which highlights rather negative data), and there is almost nothing in relation to the outcome (recovery) of individuals who complete treatment. 

However, people with SUD in residential treatment are still numerous (13,671 in Italy, 10.1% compared to those in treatment in public health services [33]) and constitute a particular population, with characteristics that can favor the treatment of many comorbidities (including HCV infection) and promote a strong motivation to change and a prioritization of health in general.

The major potential drawback of TCs, also from the HCV micro-elimination perspective, is the high risk of dropout due to drug craving or lack of motivation (“early dropout”), in addition to the emergence of deeper psychological or psychiatric problems of the person if not properly understood and treated (“late dropout”). 

The experience described, relating to a TC scan that hosts an average of 1000 people with SUD (decreased to 800 in the two years of the COVID-19 pandemic), highlights how a residential, drug-free care context is also optimal for the micro-elimination of HCV, an objective achieved in the present study. 

Overall, 68% of the eligible people started DAA treatment, 222 completed it, and the SVR rate was obtained as 100% in people for whom it could be evaluated. The remaining three subjects who were not evaluated at the standard 12 weeks after the end of treatment were assessed as being negative at the end of the treatment protocol (EoTR). Given the almost complete efficacy of a cure in our population, all 225 subjects who completed the DAA treatment can be considered cured.

HCV DAA treatment has shown to have high efficacy also in PWIDs, and the proportion of persons achieving an HCV cure is comparable with non-PWID controls [34]. The lower adherence and possibility of cure observed in the past years before DAA introduction in PWIDs treated with interferon-based therapy [35] still influence physicians and healthcare payers in deferring HCV treatment and can represent a potential barrier to DAA prescription in this population [36]. However, this presumption has not been confirmed by our and other studies [34]. In our experience, we didn’t observe any problem with adherence to HCV treatment, probably because of the strong motivation to their own physical health of the individuals who, with great effort, are recovering from addictions. Indeed, we observed a reciprocal reinforcing of motivation between addiction and HCV treatment. In other experiences, adherence by PWIDs to DAA treatment was reduced for those without stable housing [37] and increased in community health centers [38], confirming that where continuity of care can be directly monitored, as in a TC, optimal adherence can be achieved.

In analyzing reasons why not all patients with HCV infection were able to receive treatment (overall, 106 cases), two main causes can be identified: (a) treatment dropouts and/or regular discharges, and (b) bureaucratic–administrative problems. Since all RNA-positive individuals were informed and aware of their need for HCV treatment, which is freely available through the Italian National Health Service, we are confident that those (n = 62) who successfully completed the TC program and were discharged could have been treated thereafter. Considering the remaining 42 cases who dropped out, given the high probability of relapse in intravenous drug use, they remain a potential source of HCV spread among other PWIDs; however, they have also a consistent opportunity to be treated for HCV in different contexts of drug addiction care, such as public health centers delivering OAT, or low-threshold services.

Bureaucratic–administrative problems had, in our opinion, a much more consistent impact. Initially, antivirals could be prescribed only to patients with more advanced liver fibrosis (F3–F4). Nonetheless, after overcoming these limitations and extending the treatment to all chronically HCV-infected subjects irrespective of liver disease prioritization, the waiting times for the availability of drugs were still very long. Recently, the waiting times have shortened, and a rapid cascade of care has been possible, significantly decreasing the number of untreated patients from one hundred in the first two years to six in recent years (2020–2022). Still, the bureaucracy needed to start DAA treatment after subject enlistment requires 3–4 months.

Nevertheless, the risk of HCV reinfection after DAA treatment is high in active PWIDs [12] and represents one of the barriers to the goal of HCV micro-elimination in this population to reduce the risk of HCV reinfection (and infection). PWIDs should be better informed about the injecting behaviors associated with the risk of HCV transmission. For example, in our cohort, few of them had knowledge of the risk associated with the sharing of paraphernalia other than syringes (spoons, filters, etc.), or dividing doses through front-loading/back-loading (personal communication, paper in preparation).

This study was not designed to assess the outcome of the therapeutic program for addiction but, through the network of non-profit, private associations and/or direct contacts between discharged PWIDs with their educators in SPTC, we have reliable information about what happens in their daily life, including eventual drug use relapses. 

Assuming that reinfection risk is practically absent in those individuals who achieve both HCV eradication and drug addiction recovery, we can claim that 187 PWIDs (84.2%) did not experience HCV reinfection because they remained abstinent from drugs; for 134 (71.7%) of them, we also have the virological confirmation of persistent HCV elimination.

We cannot exclude that HCV reinfection could have occurred in those twenty-six PWIDs who relapsed on drug use, as well as in the remaining nine PWIDs lost to follow-up after leaving the therapeutic program. 

In conclusion, residential TCs stand as an optimal point of care for the treatment of HCV infection in PWIDs. The risk of losing patients for HCV treatment could be greatly reduced by speeding up bureaucratic processes, whereas integrating HCV and addiction treatment stands as the best prevention of reinfection with HCV.

## Figures and Tables

**Figure 1 viruses-16-00375-f001:**
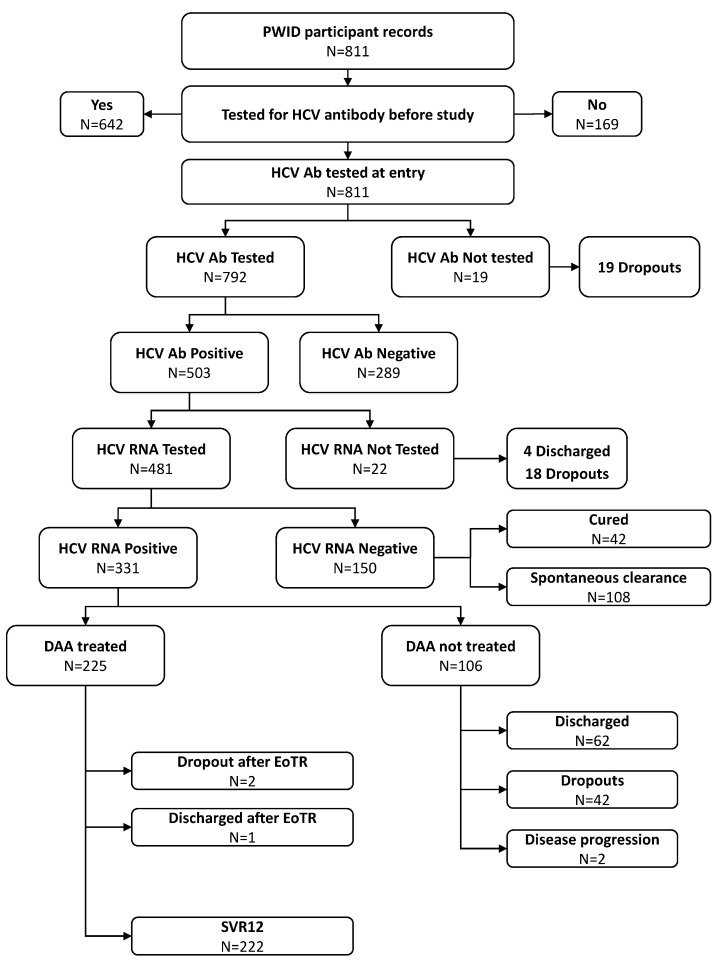
HCV screening in 811 PWIDs assisted at San Patrignano community (2018–2022) and results of the cascade of care for HCV. PWIDs: people who inject drugs; HCV: hepatitis C virus; HCV Ab: HCV antibodies; HCV RNA: HCV ribonucleic acid; DAA: direct-acting antivirals; EoTR: end of treatment response; SVR12: sustained virological response at 12 weeks after DAA end of treatment.

**Figure 2 viruses-16-00375-f002:**
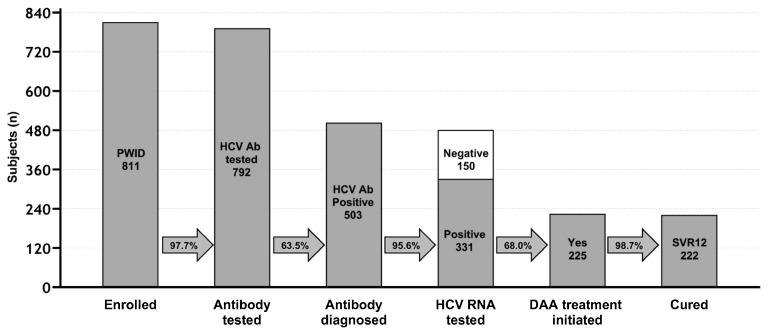
Cascade of care for HCV in PWIDs assisted at San Patrignano community (2018–2022). PWIDs: people who inject drugs; HCV: hepatitis C virus; HCV Ab: HCV antibodies; HCV RNA: HCV ribonucleic acid; DAA: direct-acting antivirals; SVR12: sustained virological response at 12 weeks after DAA end of treatment.

**Figure 3 viruses-16-00375-f003:**
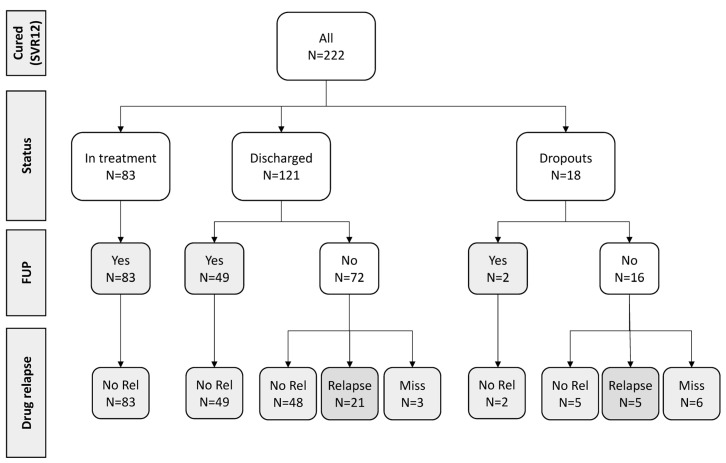
Follow-up information on 222 PWIDs cured of HCV at San Patrignano community (2018–2022). PWIDs: people who inject drugs; HCV: hepatitis C virus; FUP: follow-up; No Rel: no relapse to drug use; SVR12: sustained virological response at 12 weeks after DAA end of treatment.

**Table 1 viruses-16-00375-t001:** Characteristics of 811 PWIDs assisted at the San Patrignano community 2018–2022.

Characteristics		N (%)
All		811 (100)
Gender		
	Male	625 (77.1)
	Female	186 (22.9)
Age at start of drug injection	Median (IQR)	20 (18–24)
Age at last SPTC entry	Median (IQR)	32 (26–39)
	<30	321 (39.6)
	30–39	292 (36.0)
	40+	198 (24.4)
Year of last entry		
	<2018	409 (50.4)
	2018–2019	286 (35.3)
	2020+	116 (14.3)
Nationality		
	Italians	751 (92.6)
	Not Italians	60 (7.4)
Use of injectable substance		
	Heroin + Cocaine	740 (91.2)
	Only Heroin	49 (6.0)
	Only Cocaine	20 (2.5)
	Other	2 (0.2)
Incarceration		
	No	595 (73.4)
	Yes	216 (26.6)
HIV		
	Neg	756 (93.2)
	Pos	55 (6.8)
Previous HCV testing		
	No	169 (20.8)
	Yes, negative	265 (32.7)
	Yes, positive	377 (46.5)

Abbreviations: SPTC, San Patrignano Therapeutic Community; IQR, interquartile range; PWIDs, people who inject drugs; HIV, human immunodeficiency virus; HCV, hepatitis C virus.

**Table 2 viruses-16-00375-t002:** Characteristics of 792 PWIDs assisted at the San Patrignano community according to HCV Ab seropositivity at entry: 2018–2022.

Characteristics		Total N (%)	Neg N (%)	Pos N (%)	*p*
All		792	289 (36.5)	503 (63.5)	
Gender					0.276
	Male	609	216 (35.5)	393 (64.5)	
	Female	183	73 (39.9)	110 (60.1)	
Age at start of drug injection	Median (IQR)	32 (26–39)	29 (23–36)	35 (28–42)	<0.001 ^2^
Age at SPTC entry					<0.001 ^1^
	<30	310	154 (49.7)	156 (50.3)	
	30–39	286	99 (34.6)	187 (65.4)	
	40+	196	36 (18.4)	160 (81.6)	
Years of addiction					<0.001 ^1^
	0–4	279	151 (54.1)	128 (45.9)	
	5–9	167	69 (41.3)	98 (58.7)	
	10+	346	69 (19.9)	277 (80.1)	
Year of last entry					<0.001 ^1^
	<2018	408	120 (29.4)	288 (70.6)	
	2018–2019	275	110 (40.0)	165 (60.0)	
	2020+	109	59 (54.1)	50 (45.9)	
Nationality					0.732
	Italians	735	267 (36.3)	468 (63.7)	
	Not Italians	57	22 (38.6)	35 (61.4)	
Incarceration					<0.001
	No	580	240 (41.4)	340 (58.6)	
	Yes	212	49 (23.1)	163 (76.9)	
HIV					<0.001
	Neg	737	284 (38.5)	453 (61.5)	
	Pos	55	5 (9.1)	50 (90.9)	
Previous HCV testing					<0.001
	No	156	65 (41.7)	91 (58.3)	
	Yes, negative	259	224 (86.5)	35 (13.5)	
	Yes, positive	377	-	377 (100)	

^1^ Chi-square for trend test; ^2^ Mann–Whitney test; Abbreviations: SPTC, San Patrignano Therapeutic Community; PWIDs, people who inject drugs; HIV, human immunodeficiency virus; HCV, hepatitis C virus; IQR, interquartile range.

**Table 3 viruses-16-00375-t003:** DAA treatment initiation among 331 HCV RNA+ PWIDs assisted at San Patrignano community and analysis of factors associated: 2018–2022.

Characteristics	Total N	DAA Treatment N (%)	IR	IRR	95% CI	*p*	aIRR	95% CI	*p*
All	331	225 (68.0)	2.04						
Gender						0.766			0.448
Male	260	168 (64.6)	2.02	1			1		
Female	71	57 (80.3)	2.12	1.05	0.78–1.41		1.13	0.83–1.54	
Age (by 10-year increase)				1.08	0.95–1.24	0.241	0.93	0.79–1.09	0.360
Year of last entry						<0.001			<0.001
<2018	202	121 (59.9)	1.57	1			1		
2018–2019	96	75 (78.1)	2.59	1.65	1.23–2.20		1.48	1.09–1.99	
2020+	33	29 (87.9)	7.06	4.49	2.99–6.73		4.25	2.81–6.44	
Nationality						0.034			0.080
Italians	312	212 (67.9)	2.14	1			1		
Non-Italians	19	13 (68.4)	1.17	0.55	0.31–0.96		0.60	0.34–1.06	
METAVIR score						<0.001			<0.001
F0–F1	208	133 (63.9)	1.69	1			1		
F2	74	61 (82.4)	2.66	1.57	1.16–2.13		1.58	1.15–2.16	
F3–F4	26	22 (84.6)	4.65	2.75	1.75–4.31		2.43	1.50–3.95	
Missing	23	9 (39.1)	2.37	1.40	0.71–2.75		1.16	0.58–2.29	
HIV						<0.001			<0.001
Neg	302	201 (66.6)	1.90	1			1		
Pos	29	24 (82.8)	5.59	2.94	1.93–4.49		2.39	1.43–4.00	

Abbreviations: IR, incidence rates (×10^3^ person-days, PDs); IRR, unadjusted incidence rate ratios; aIRR, adjusted IRR (adjusted for all shown variables); DAA, direct-acting antivirals; PWIDs, people who inject drugs; HIV, human immunodeficiency virus; HCV, hepatitis C virus.

## Data Availability

The dataset will be made available upon reasonable request.

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
