# Peer review of "Integration of Hepatitis C and Addiction Treatment in People Who Inject Drugs: The San Patrignano HCV-Free and Drug-Free Experience"

_viruses, 2024, doi:10.3390/v16030375_

Round 1

Reviewer 1 Report

Comments and Suggestions for Authors

The main question addressed by the research is will the establishing of  the Cascade of Care model in the therapeutic communities be adequate for implementation of  HCV micro-elimination and reinfection prevention, based on cohort of persons who inject drugs (who represent an important reservoir HCV infection).

The HCV Cascade of Care in general is already an established model (screening, diagnosis, and care linkage to treatment initiation, completion, and the achievement of HCV cure). However the aforementioned  steps could differ.  The manuscript brings novelity in detailed description of the elements of the cascade. Additionaly, the originality of this manuscript lies in fact that there is liitle data about Cascade of Care  among San Patrignano therapeutic community which possesses therapeutic program based on education and rehabilitation which varies on every individual, so there is big importance related to th HCV micro-elimination of this TC.

The gaps of  the paper addresses are those found  along the cascade of HCV care, including lack of testing; linkage to care, knowledge-related treatment. 

Besides calculating frequencies of infection diagnosis, confirmation, treatment and achievement of a SVR, the manuscript evaluates the risk of reinfection of PWIDs achieving HCV eradication. The risk of reinfection is a very important part of the CoC, and it was determined by collecting follow-up virologic information of previously recovered individuals and eventual relapse in drug use, assuming the latter as a potential source of reinfection.

Given that it was written in manuscript  that 106 subjects (32.0%) HCV-RNA positive PWIDs did not receive DAA therapy due to either drop-out (n=42 - 39.6%) or being discharged from SPTC before the start 240 of  DAA treatment (n=62 - 58.5%) do you think that this is also a significant group in terms of reinfection and spread of infection and do you find in the literature some measures proposal (references) or you suggest them yourself, so so they could be converted into special Cascade of Care ?

The conclusions are consistent with the evidence because it is evident that exactly established CoC steps were used, and the procedure for including or excluding patients from the analysis and further statistical processing was described exactly, which resulted in accurate conclusions. However, the ultimate accuracy of these conclusions will be verified in practice, in terms of the degree of success of HCV elimination using this CoC model.

All main questions  posed were addressed and by describing all the elements of the CoC – testing all the individuals on the admission, determining seropositivity i.e. antibodies against HCV,  then the PCR HCV-RNA testing, analyzing the group achieving SVR after DAA treatment, and analyzing the group on follow-up which remained HCV-free.

The manuscript  results indicate that enhancing PWID's access to community-based models of care, retention in the HCV care cascade and education regarding DAA therapy will help HCV elimination.

The references are appropriate. The quality of the data presented on the tables is adequate , and qualitatively statistically processed and in concordance with the text of the manuscript.

Author Response

We would like to thank the reviewer for the remarks and suggestion made. We have modified the paper according to this review and we have attached a new version of the manuscript in two versions. In the first version we highlighted in cyan and barred those part that were modified/erased, while in yellow the new part added (viruses-2855693_Rev_feb24_highlighted.docx). The second one (viruses-2855693_Rev_feb24.docx) contain the final version of the manuscript. Figure 1 and Figure 3 were modified only removing colours.

Please find in detail the answer to the single Reviewer.

Reviewer 1

#1

Given that it was written in manuscript  that 106 subjects (32.0%) HCV-RNA positive PWIDs did not receive DAA therapy due to either drop-out (n=42 - 39.6%) or being discharged from SPTC before the start of DAA treatment (n=62 - 58.5%) do you think that this is also a significant group in terms of reinfection and spread of infection and do you find in the literature some measures proposal (references) or you suggest them yourself, so they could be converted into special Cascade of Care ?

Response: We thank the reviewer for the comment. We further discuss this point in the Discussion (lines 384-394 in the highlighted version of the revised manuscript)

#2

The manuscript results indicate that enhancing PWID's access to community-based models of care, retention in the HCV care cascade and education regarding DAA therapy will help HCV elimination.

Response: We thank the reviewer for the comment. We stressed this consideration adding a paragraph and five new references (lines 371-384 in the highlighted version of the revised manuscript)

Reviewer 2 Report

Comments and Suggestions for Authors

The authors describe the cascade of care of HCV treatment in the therapeutic community of San Patrugnano, Italy. The study is informative and well-designed, and the manuscript is well-written. I only have minor comments that the authors may consider.

1. Line 22-24: this very long sentence is hard to follow. Consider dividing it.

2. Line 93: note that the cascade of care is already abbreviated as CoC in line 50, and the same for the therapeutic community in line 80.

3. It is necessary to mention that a total of 811 PWIDs were admitted at SPTC in the abstract.

4. Did the authors sequence the 331 HCV RNA-positive specimens?

5. The authors should discuss and compare with more relevant studies conducted previously and elsewhere. Very few references were cited in the discussion section.

Author Response

We would like to thank the reviewer for the remarks and suggestion made. We have modified the paper according to this review and we have attached a new version of the manuscript in two versions. In the first version we highlighted in cyan and barred those part that were modified/erased, while in yellow the new part added (viruses-2855693_Rev_feb24_highlighted.docx). The second one (viruses-2855693_Rev_feb24.docx) contain the final version of the manuscript. Figure 1 and Figure 3 were modified only removing colours.

Please find in detail the answer to the single Reviewer.

Reviewer 2

#1

Line 22-24: this very long sentence is hard to follow. Consider dividing it.

Response: we divided the sentence

#2

Line 93: note that the cascade of care is already abbreviated as CoC in line 50, and the same for the therapeutic community in line 80.

Response: we have modified as suggested.

#3

It is necessary to mention that a total of 811 PWIDs were admitted at SPTC in the abstract.

Response: We have mentioned the number of PWIDs admitted at SPTC in the abstract (Line 28 in the highlighted version).

#4

Did the authors sequence the 331 HCV RNA-positive specimens?

Response: Thanks for the suggestion. All HCV-RNA positive specimens were sequenced, and results were added in a paragraph in the Results section (line 241-244 in the highlighted version).

#5

The authors should discuss and compare with more relevant studies conducted previously and elsewhere. Very few references were cited in the discussion section.

Response: We appreciate for the suggestion, and we have added two paragraphs in the discussion (lines 371-395 and 405-411 in the highlighted version) with 5 more references (34-38). We have added the references in the discussion section of the manuscript.

Reviewer 3 Report

Comments and Suggestions for Authors

To improve the article, I have a couple of suggestions for the authors:

1.      Please, in Table 2, do not state only previous testing but also previous antiviral treatment and mention how many patients entering anti-HCV therapy were “reinfections”. These data would be of paramount importance from the epidemiological point of view.

2.      Discuss the risk of reinfection in the discussion section and approaches to prevent reinfection. The risk of reinfection represents one of the arguments against PWID therapy.

3.      Focus also on measures to improve adherence to therapy in the discussion: the previously published studies discuss stable housing as an important adherence factor (Frankova S et al., 2021, Harm Reduction Journal); other factors are discussed in the HERO study (Lopes S.S. et al., International Journal of Drug Policy, 2024). 

Author Response

We would like to thank the reviewer for the remarks and suggestion made. We have modified the paper according to this review and we have attached a new version of the manuscript in two versions. In the first version we highlighted in cyan and barred those part that were modified/erased, while in yellow the new part added (viruses-2855693_Rev_feb24_highlighted.docx). The second one (viruses-2855693_Rev_feb24.docx) contain the final version of the manuscript. Figure 1 and Figure 3 were modified only removing colours.

Please find in detail the answer to the single Reviewer.

Reviewer 3

To improve the article, I have a couple of suggestions for the authors:

#1

Please, in Table 2, do not state only previous testing but also previous antiviral treatment and mention how many patients entering anti-HCV therapy were “reinfections”. These data would be of paramount importance from the epidemiological point of view.

Response: We thank the reviewer for the comment. We modified two paragraphs in the results adding more details on previous testing and treatment (lines 198-204 in the highlighted version) and about reinfection (lines 245-247 in the highlighted version).

#2

Discuss the risk of reinfection in the discussion section and approaches to prevent reinfection. The risk of reinfection represents one of the arguments against PWID therapy.

Response: As suggested, we have added three paragraphs in the Discussion dealing with adherence and risk of reinfection in PWID (Lines 371-395 and 405-411 in the highlighted version).

#3

Focus also on measures to improve adherence to therapy in the discussion: the previously published studies discuss stable housing as an important adherence factor (Frankova S et al., 2021, Harm Reduction Journal); other factors are discussed in the HERO study (Lopes S.S. et al., International Journal of Drug Policy, 2024). 

Response: We thank for the suggestion. As mentioned above (#2), we have expanded the discussion adding five more references (including the suggested references) to the Discussion section of the manuscript (Lines 371-395 and 405-411 in the highlighted version).
